# Asynchronous Communication in Spectral Element and Discontinuous Galerkin Methods for Atmospheric Dynamics – A Case Study Using the Higher Order Methods Modeling Environment (HOMME-homme_dg_branch)

Benjamin F. Jamroz[1] and  Robert Klöfkorn[1,2]

[1] Computational Information Systems Laboratory, National Center for Atmospheric Research , 1850 Table Mesa Drive, Boulder, CO 80305
[2] International Research Institute of Stavanger, P. O. Box 8046, 4068 Stavanger, Norway

**Abstract.** The scalability of computational applications on current and next generation supercomputers is increasingly limited by the cost of inter-process communication. We implement non-blocking asynchronous communication in the High-Order Methods Modeling Environment for the time-integration of the hydrostatic fluid equations using both the Spectral Element and Discontinuous Galerkin methods. This allows the overlap of computation with communication effectively hiding some of the costs of communication. A novel detail about our approach is that it provides some data movement to be performed during the asynchronous communication even in the absence of other computations. This method produces significant performance and scalability gains in large-scale simulations.

**Keywords.**  Asynchronous communication, Spectral Element, Discontinuous Galerkin, CAM, HOMME

## 1   Introduction

The Community Earth System Model (CESM) is a global climate model with full coupling between the atmosphere, ocean, land, sea-ice, and land-ice components (Gent et al. (2011)). The Community Atmosphere Model (CAM) is the atmospheric component in CESM which advances the physical attributes of the atmosphere as well as time-integrating the atmospheric dynamics through the use of a dynamical core (Neale et al. (2010)). Although there are several dynamical cores available in CAM, the High-Order Methods Modeling Environment (HOMME) dynamical core (Dennis et al. (2012)) is most widely used for large-scale simulations on supercomputers due to its scalability.

HOMME was originally designed and optimized for the spectral element (SE) method but nowadays also supports a discontinuous Galerkin (DG) approach to advance the hydrostatic primitive equations. Both methods have been chosen for their scalability on large distributed memory supercomputers. The high-order of accuracy of these methods is complemented with a compact communication pattern between representative elements. Specifically, in two-dimensions each element needs only to exchange information with its edge neighbors (DG), or edge and vertex neighbors (SE). Unlike a finite-volume method where higher-order stencils have larger spatial extent, the SE and DG methods attain this property for arbitrary order, at the expense

of a smaller timestep. These schemes limit the amount of inter-process communication, providing superior scalability in many applications.

HOMME has demonstrated very good scaling for both the SE and DG methods. The SE method has shown good scaling up to 178k cores (Dennis et al. (2012)), while the DG method has shown similar scaling beyond 2k cores (Nair et al. (2009)) and very recently up to 16k cores (Nair et al. (2016)) and up to 21k cores as part of this work. Excellent scalability of both methods, SE and DG, on leadership class supercomputers is also reported in other implementations, for example, in the Nonhydrostatic Unified Model of the Atmosphere (NUMA, Müller et al. (2015)). Although HOMME scales well, further improvements in performance and scalability can increase the amount of simulated years of climate per day (SYPD) of CESM on large parallel resources. This reduces the time required for long simulations and increases the amount of science obtained in a given amount of wall-clock time. Additionally, better scalability yields more efficient use of large-scale computational resources. Even a small reduction of computational time can have a large impact in reducing the operational costs of a large supercomputer. Finally, next-generation hardware, which is typically characterized by lower clock frequencies and less memory per core, will benefit from additional parallelism, concurrency, and asynchronicity as pointed out in (Keyes (2011)).

In this paper, we discuss the implementation of non-blocking asynchronous communication in HOMME for both the SE and DG methods. We highlight that our method provides some data movement to be performed, even in the absence of additional computation, during the communication step. Overlapping communication with this data movement and additional computation shows scalability and performance gains on large-scale simulations. To our best knowledge this has not been published before. Contemporary works discussing scalability of dynamical cores for climate and weather prediction on leadership class supercomputers such as NUMA (Müller et al. (2015)) or the Icosahedral Nonhydrostatic model (ICON) and the Model for Prediction Across Scales – Atmosphere (MPAS-A) (both in Brömmel et al. (2015)) do not mention asynchronous communication. Only the Nonhydrostatic Icosahedral Atmospheric Model (NICAM, Kodama et al. (2014)) employs asynchronous communication but overlapping communication with computation is not presented. Besides these works, a vast number of papers focusing on the scalability of simulation software on supercomputers also exists. Some state-of-the-art works have been presented at the *International Conference on High Performance Computing, Networking, Storage and Analysis* (SC) conference series (cf. Chhugani et al. (2012); Bermejo-Moreno et al. (2013); Heinecke et al. (2014); Rudi et al. (2015)) and special extreme scaling workshops (cf. Brömmel et al. (2015, 2016)). All of these works mention the usage of asynchronous communication but do not provide algorithmic or implementation details. Additionally, the work of Wittmann et al. (2013) that entirely focuses on the topic of asynchronous communication does not provide any details on how to incorporate asynchronous communication overlappping with computation in a simulation code. To that end we are not aware of any work providing these details.

The outline of this paper is as follows. First, we present the existing data structures and communication strategy in HOMME. Next, we summarize our implementation of non-blocking asynchronous communication highlighting data movement which can be performed during communication. We then present scaling results and discuss advantages and limitations of the new method.

## 2 Background

We first give some background on non-blocking message passing using Message Passing Interface (MPI) (Forum (1994)). Next, in order to clearly explain the non-blocking asynchronous communication method we first describe the data structures used in HOMME and the existing synchronous communication method.

### 2.1 Non-blocking Communication

Many high-performance scientific applications use MPI to communicate between processes in a distributed memory context. Point-to-point messaging is one of the communication paradigms implemented by MPI, others include reductions, broadcasts, scatters, and gathers. This communication method is often used in the context of nearest neighbor communication in the solution of partial differential equations using explicit in time integration methods where data between neighboring grid elements (finite volume cells, Galerkin elements) must be exchanged. Point-to-point messaging is characterized by one process (the "sender") sending data to another (the "receiver").

Blocking communication is used when the MPI processes involved cannot advance during a point-to-point communication cycle. Here, a process sending a blocking message, typically using a call to MPI_Send, must wait until the message has been sent and the storage buffer is ready to be reused. Likewise, in a blocking receive, using MPI_Recv, the receiver must wait for the message to be fully received. Since blocking communication effectively causes a synchronization between the sending and receiving processes this method is not widely used in high-performance parallel applications.

A non-blocking implementation allows sending messages without the restriction that the sending process waits for the message to be received. On the receiver side, the destination process posts a receive, but can continue running without waiting for the message to be received. Thus, both the source and destination processes can continue execution while the message is sent and received. This allows the overlap of some computation during communications, giving the potential to hide some of the cost of communication. In most applications, however, there is a point in the calculation at which the message needs to be fully sent and received before any more progress can be made. At this point, the receiver must wait for the message to be completely received and the sending process must wait for the send to be fully completed. Most commonly, non-blocking communication is implemented using MPI with the MPI_Isend, MPI_Irecv, and MPI_Wait/MPI_Waitall calls.

The effectiveness of non-blocking communication depends on the MPI library implementation (cf. Wittmann et al. (2013)) but also on system specific characteristics which are not fully encapsulated in the MPI layer. A measure of the effectiveness of non-blocking communication is provided by the MPI_overhead test as a part of the Sandia MPI Micro-Benchmark Suite (San). Here, non-blocking communication between two processes is initialized using MPI_Isend and MPI_Irecv. Then some computation is performed before a call to MPI_Waitall. The amount of computation is increased in each iteration, and each phase is timed to find the point at which the computation costs dominate the non-blocking communication costs. The benchmark then reports a metric for what percentage of the time can be used for computation for a given message size. We used this benchmark to investigate the performance of two different MPI implementations, IBM's version of MPICH 1.5 and Intel MPI version 4.0.3.008, and different run time parameters (i.e. environment variables) on the Yellowstone supercomputer (Yel).

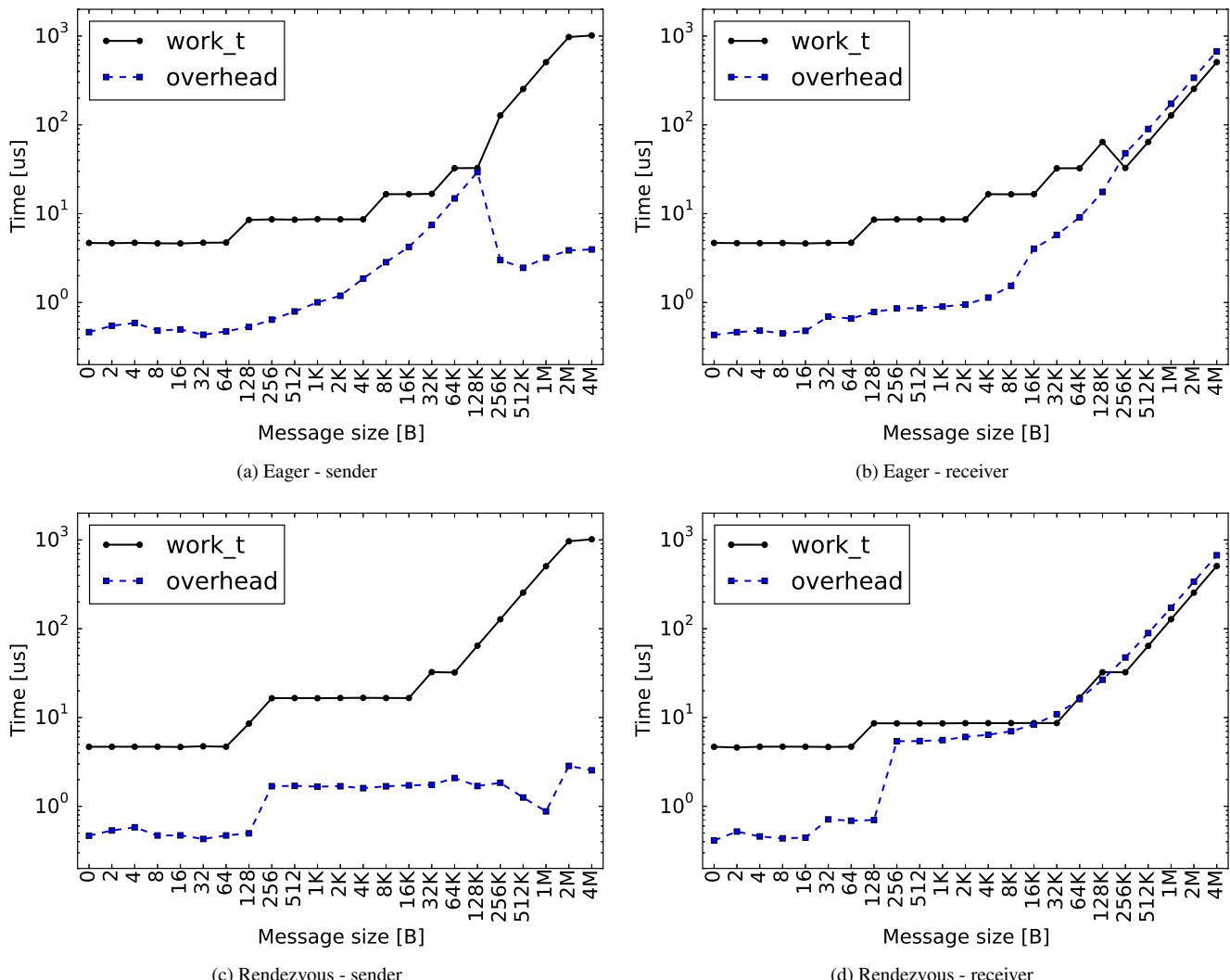

(a) Eager - sender

(b) Eager - receiver

(c) Rendezvous - sender

(d) Rendezvous - receiver

**Figure 1.** Results from the Sandia MPI MicroBenchmark using the IBM MPI implementation with Eager protocol (top) for sending (a) and receiving (b) an asynchronous non-blocking message and Rendezvous protocol (bottom) for sending (c) and receiving (d). Here, overhead corresponds to the amount of overhead time required to send or receive a non-blocking message, while work_t corresponds to the amount of computation required to effectively hide the costs of sending or receiving the message.

Figure 1 shows the results of the micro-benchmark for various message sizes sent between two nodes of the Yellowstone supercomputer (Yel) averaged over 100 iterations for both the Eager and Rendezvous protocols using the IBM MPI implementation. Figure 1 (a) shows the overhead and work_t metrics for sending a non-blocking message for this micro-benchmark using the Eager protocol. Here, overhead signifies the time spent sending the non-blocking message, while work_t denotes the amount of computational time estimated to fully hide the resulting cost of waiting for the message being received. Similarly,

Figure 1 (b) shows the same data for the receiver's side. Figures 1 (c-d) show similar results for the Rendezvous protocol. In these plots, we can see that the overhead of asynchronous non-blocking messaging increases with message size. Additionally the amount of overlapped computation required to effectively hide the cost of communication increases with message size. This shows that in order to effectively hide communication costs using asynchronous non-blocking communication, one must provide enough computation to be performed during the communication step. Providing only a small amount of computation to be performed during communication limits the benefit of non-blocking asynchronous communication.

## 2.2 Current Communication Strategy

The computational grid in HOMME is typically a semi-structured cubed-sphere or fully unstructured static grid on the surface of a sphere. Due to the time-scale separation of hydrostatic flows in the locally horizontal (along the surface of the sphere) and locally vertical (radial) directions, only the surface of the sphere is discretized using the SE or DG methods. The vertical direction uses centered finite-difference methods (Simmons and Burridge (1981)). This effectively creates a stack of elements, an "element-column", with one two-dimensional element for each vertical level. Typically, for climate simulations, there are 26-50 vertical levels, although some whole-atmosphere models consider up to 81 levels (Liu et al. (2010)). For parallel efficiency all vertical levels, one element-column, exist on the same process.

In integrating the dynamics of the hydrostatic equations, the majority of the computations are within each element at one level. Additionally, the consistency conditions between elements (continuity for SE, flux-balance for DG) only involves horizontally adjacent neighboring elements at the same vertical level. Thus the layout of the element data in HOMME has the form

```
type element
    real, dimension(np, np, nlev) :: element_data
end type element
```

where $np$ represents the number of Gauss-Lobatto Lebesgue (GLL) points, and equivalently $np-1$ denotes the order of polynomial, and $nlev$ denotes the number of vertical levels. Since the data within one element (at one vertical level) is co-located with stride one access, intra-element computations, which represent the bulk of the computation, can be done with maximal efficiency. However, at certain points in the calculation, e.g. when calculating the surface pressure, a reduction across vertical levels must be performed. Although this data structure is not ideal for this particular calculation, it is a small percentage of the overall computation. Thus the above data structure is optimal for the majority of calculations.

Consistency between neighboring elements is one place where communication between elements, and therefore processes, must occur. In HOMME, for both the SE and the DG methods this amounts to exchanging data between neighboring horizontal elements. For the SE method, since continuity must be enforced, the horizontal neighbors with which information must be exchanged include elements which share an edge and those which only share a vertex. For the DG method, since only edge fluxes between elements is required, only the neighboring elements which share an edge are included. Figure 2 illustrates the connectivity of a reference element for the SE and DG methods.

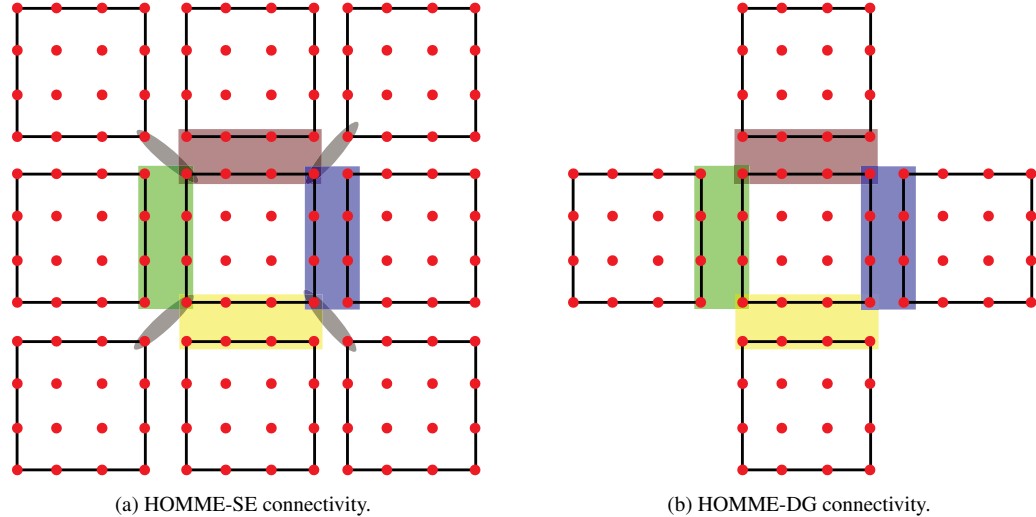

(a) HOMME-SE connectivity.        (b) HOMME-DG connectivity.

**Figure 2.** Connectivity in HOMME-SE (a) and HOMME-DG (b). The DG version does not need to communicate vertex data and thus connectivity to other processes is reduced.

The existing communication method for both the SE and DG methods has the following form. First, the element data, which is represented above with a three-dimensional index, is packed into a one-dimensional buffer consistent to what is required by the calls to MPI_Irecv, MPI_Isend. The packing takes all of the edge and vertex values and writes them into a buffer in a co-located manner. Once all of the data for each element-column on a process has been packed into the buffer, the appropriate

MPI_Irecv and MPI_Isend calls are made. Immediately after all of these calls have been made, a call to MPI_Waitall is called on all of the receive and send requests. After this point, the data can be additively unpacked from the buffer into the element data structures. Although this communication pattern is technically asynchronous (because of the use of MPI_Irecv and MPI_Isend) the immediate use of MPI_Waitall creates a synchronization across processes and we therefore denote this communication pattern as synchronous. In runs on large numbers of processes, there is a significant amount of time spent at

this call where processes wait for neighboring processes to both send and receive data.

## 3 Overlapping Asynchronous Communication Strategy

In order to implement effective non-blocking asynchronous communication in HOMME we have revised the communication pattern. In the existing implementation, element edges and vertices are packed (unpacked) into (out of) a buffer sequentially, in order of element index, with no regard for whether the data needs to be messaged. This is a key distinction from our method

which takes into account this information. Here, we have separated the packing and unpacking of element edges and vertices into groups corresponding to individual messages to be sent and received. This modification allows us to overlap the packing

and unpacking of edges with the communication. This approach also provides the ability to perform some data movement even in the absence of any other computation. We now describe this technique.

To implement the overlap of pack/unpack routines with the communication itself we generated the following mapping. Denote by $\mathcal{L}_p$ the set of all processes with which a given process needs to communicate. Using this set we generate a set of elements that contains all elements $e \in \mathcal{E}_l$ that are linked to process $l \in \mathcal{L}_p$, either the edge or the vertices (see also Figure 2). This latter set specifies the data that needs to be packed before message $l$ is sent. Specifically, after packing all of the edges and vertices for message $l$ one can immediately call MPI_Isend, and begin packing the data for the next message.

On the receive side, one can unpack data as soon as a message is received. Specifically, we use a call to MPI_Testany to determine if any of the messages have been received. After a message has been received, we remove it from the list of messages to be checked in MPI_Testany and unpack the data that was received. We repeat this process with a reduced list of messages in the call to MPI_Testany until all of the messages have been received and the corresponding data has been unpacked. Note that in general the connectivity for send and receive could differ, i.e. we have a set $\mathcal{L}_p^s$ for the send procedure and $\mathcal{L}_p^r$ for receive. However, the communication we consider in this paper is symmetric, i.e. $\mathcal{L}_p^s = \mathcal{L}_p^r$.

In Algorithm 1 we present the **packAndSend** routine and in Algorithm 2 the **receiveAndUnpack** routine. Both overlap the send/receive with the corresponding pack/unpack.

---
**Algorithm 1** packAndSend
---
1: **MPI_Waitall**( $\mathcal{L}_p^s$ ) { wait for previously posted MPI_Isend calls }

2: **for** $q \in \mathcal{L}_p^s$ **do**

3:     **for** $e \in \mathcal{E}_q$ **do**

4:         **packData**( $e, q$ ) { pack data to MPI message buffer }

5:     **end for**

6:     **MPI_Isend**( $q$ ) { send data in message buffer to rank $q$ }

7: **end for**

---

Most notable about the implementation explained above is that even in the absence of additional computation to be completed during communication, the packing and unpacking of the buffers provides some data movement to be accomplished while waiting for messages to be received. This is extended in the case where there are multiple elements per process. Here, these intra-process edges and vertex contributions are packed and unpacked in between the send and receive stages, providing even further data movement before querying for completed messages. More internal edges and vertices provide more data movement and therefore better communication hiding.

Finally, since our communication restructuring now clearly supports separate send and receive routines, one can now place computation between these calls to potentially hide even more of the communication costs. In many cases, however, this requires some algorithmic restructuring which is not always easy or possible. For that reason our implementation provides at least the more simple overlap of pack/unpack with communication calls. It is important to mention that this approach can in principle be applied to any point-to-point communication stemming from the discretization of PDEs or other applications one

**Algorithm 2** receiveAndUnpack

---

1: $n_r \leftarrow 0$

2: **while** $n_r < |\mathcal{L}_p^r|$ **do**

3:    { check if message is available, if yes then $q$ contains the corresponding rank}

4:    **if** **MPI_Testany**( $\mathcal{L}_p^r, q$ ) **then**

5:       **for** $e \in \mathcal{E}_q$ **do**

6:          **unpackData**( $e, q$ ) { unpack data from MPI message buffer }

7:       **end for**

8:       reset **MPI_Request** for $q$ to **MPI_REQUEST_NULL**

9:       $n_r \leftarrow n_r + 1$ { increase received counter }

10:    **end if**

11: **end while**

---

example being the semi-Lagrangian multi-tracer transport schemes implemented in HOMME by Erath et al. (2012) and Erath and Nair (2014). We now describe the computation and data movement that can be performed while waiting for messages to be received in the SE and DG methods.

### 3.1 Overlapping for the SE method

In the SE method, communication is required mainly as part of an operator which projects data for each element (which is redundant a the edges of the element) onto the space of continuous piecewise polynomials (Taylor and Fournier (2010)). Specifically, data on element edges is not continuous until after a pack, communication, unpack cycle has been completed. This adds a difficulty in overlapping computation with communication for the SE method since any computation depending upon the data being messaged would have to take into account the discontinuity of the data.

While we haven't been able to take advantage of any significant computation to be performed while communication occurs, there is still the data movement performed by the packing and unpacking of interior data and the packing and unpacking of messages as they arrive. Since this data movement is required in the original synchronous communication method as well, overlapping this data movement provides a small amount of work to be done to hide some of the communication costs.

### 3.2 Overlapping for the DG method

In the DG method, communication is required to obtain data needed to perform flux calculations carried out at each edge of an element (Nair et al. (2009)). This allows the computation of internal edge and element integrals during the asynchronous communication. We have allowed the computation of auxiliary diagnostic variables between the call of send and receive. Further code revision could include the computation of the right hand side and internal flux computations as described in (Baggag et al. (1999)). In Algorithm 3 we describe how we overlap the computation of auxiliary variables and the computation

of the gradient of the solution for the diffusion operator with the communication of the fluxes. Details on the implementation of the diffusion operator can be found in (Nair (2009)).

---

**Algorithm 3** dg3d_uv_step

---

1:  **dg3d_packAndSend**( userdata ) {send data for flux and gradient computation}

2:  **gradient_p3d**( userdata ) {compute local auxiliary variables }

3:  **dg3d_recvAndUnpack**( userdata ) {receive data}

4:  **if** updateDiffusion **then**

5:      **dg3d_diff_grads_uv**( userdata ) {compute local gradients}

6:      **dg3d_gradientPackAndSend**( userdata )

7:  **end if**

8:  rhs ← **dg3d_uvform_rhs** {compute fluxes and right hand side}

9:  **if** updateDiffusion **then**

10:     **dg3d_gradientRecvAndUnpack**( userdata ) {receive the gradients}

11:     diff_rhs ← **dg3d_diff_flux**( userdata ) {compute gradients fluxes}

12: **end if**

13: **if** diffusion **then**

14:     **rhs = rhs + diff_rhs**

15: **end if**

---

In addition, in comparison to the DG implementation used in (Nair et al. (2009)) which uses the same communication structure as the SE method (which means unnecessary communication of vertex values) the new DG implementation only communicates edge values (see Figure 2b). This is easily achieved by simply altering the sets $\mathcal{L}_p^s$ and $\mathcal{L}_p^r$. This reduces the inter-process connectivity considerably. The result is faster execution times and better scaling as presented in the next section.

## 4 Results

We test our implementation of non-blocking asynchronous communication using the well known Jablonowski-Williamson baroclinic wave instability test case (Jablonowski and Williamson (2006)) using the Yellowstone supercomputer (Yel). We first show that the new communication strategy allows us to reproduce the results obtained with the pre-existing communication strategy. Then we show results for strong scalings on representative climate simulation resolutions. For all of the following runs we have used a cubed-sphere grid with $n_e$ elements along each edge of the cube for a total of $E \equiv 6n_e^2$ total elements.

### 4.1 The Jablonowski-Williamson baroclinic wave instability test case

The Jablonowski-Williamson baroclinic wave instability test case examines the evolution of an idealized baroclinic wave in the northern hemisphere. This test is designed to evaluate dynamical cores at resolutions applicable to climate simulations. Thus, it

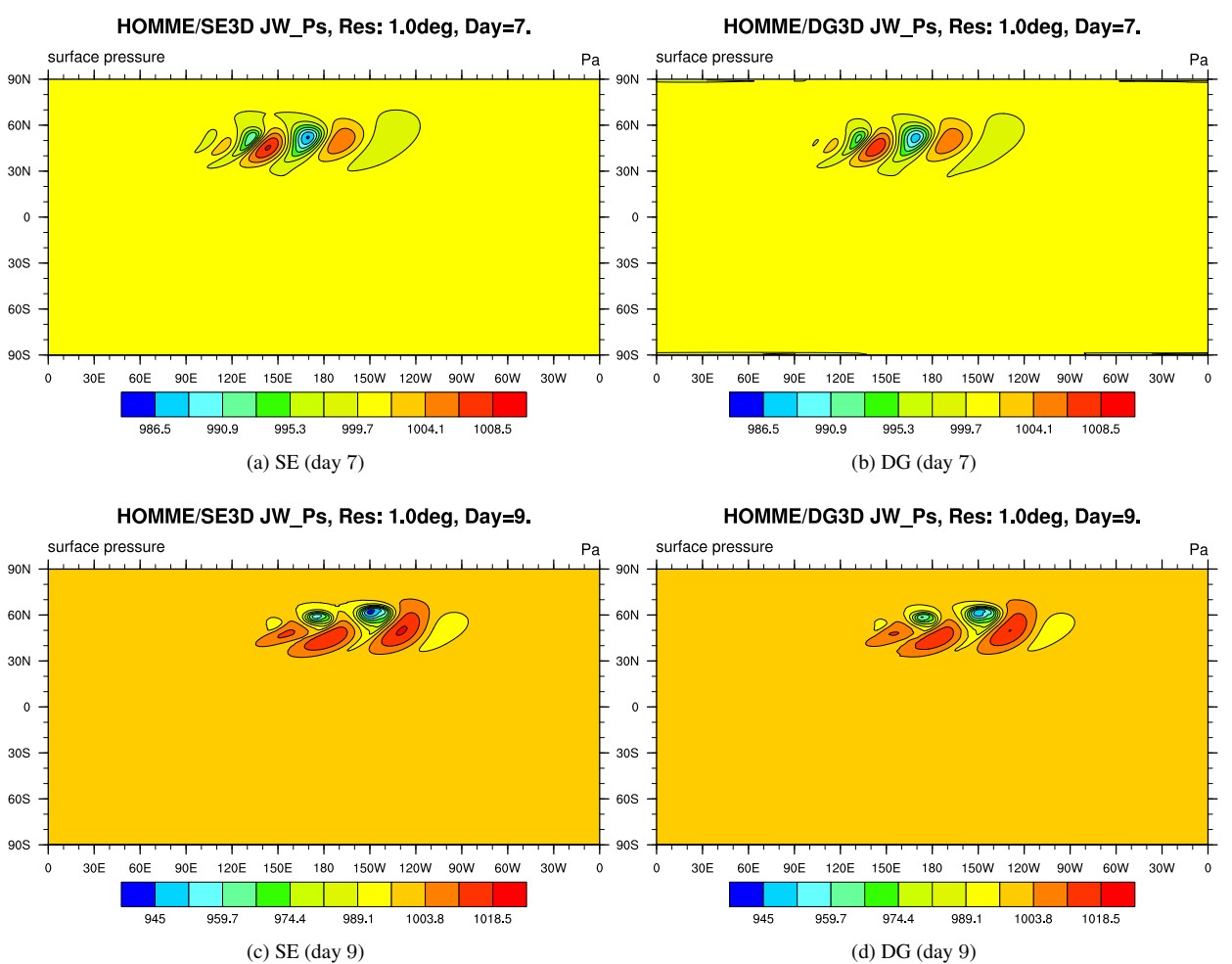

**Figure 3.** Surface pressure at day 7 and 9 for the HOMME-SE (a,c) and HOMME-DG (b,d) code for the Jablonowski-Williamson baroclinic wave instability test case. Both methods used 1 degree resolution at the equator ($nlev = 26$, SE: $np = 4$, $n_e = 30$, DG: $np = 6$, $n_e = 18$).

is a good case to get a measure of performance and scalability in a climate realistic test problem. Although an analytic solution is not available for this test case, reference solutions exist for the Eulerian dynamical core (Neale et al. (2010)).

In Figure 3 and 4 we present the results for the surface pressure and the vorticity, respectively, for the Jablonowski-Williamson test case (dcm; Jablonowski and Williamson (2006)) using non-blocking asynchronous communication. We run both methods, the SE and the DG, for this test case using a resolution of roughly 1 degree at the equator. For the SE method this means $n_e = 30$, since we are using the standard configuration of $np = 4$. For the DG method, we use $np = 6$ and $n_e = 18$. Both models use $nlev = 26$. As Figure 3 and 4 show, both methods we are able to reproduce the results presented in the literature (dcm; Jablonowski and Williamson (2006)). For the DG method we are able to achieve bit-for-bit reproducibility of the results

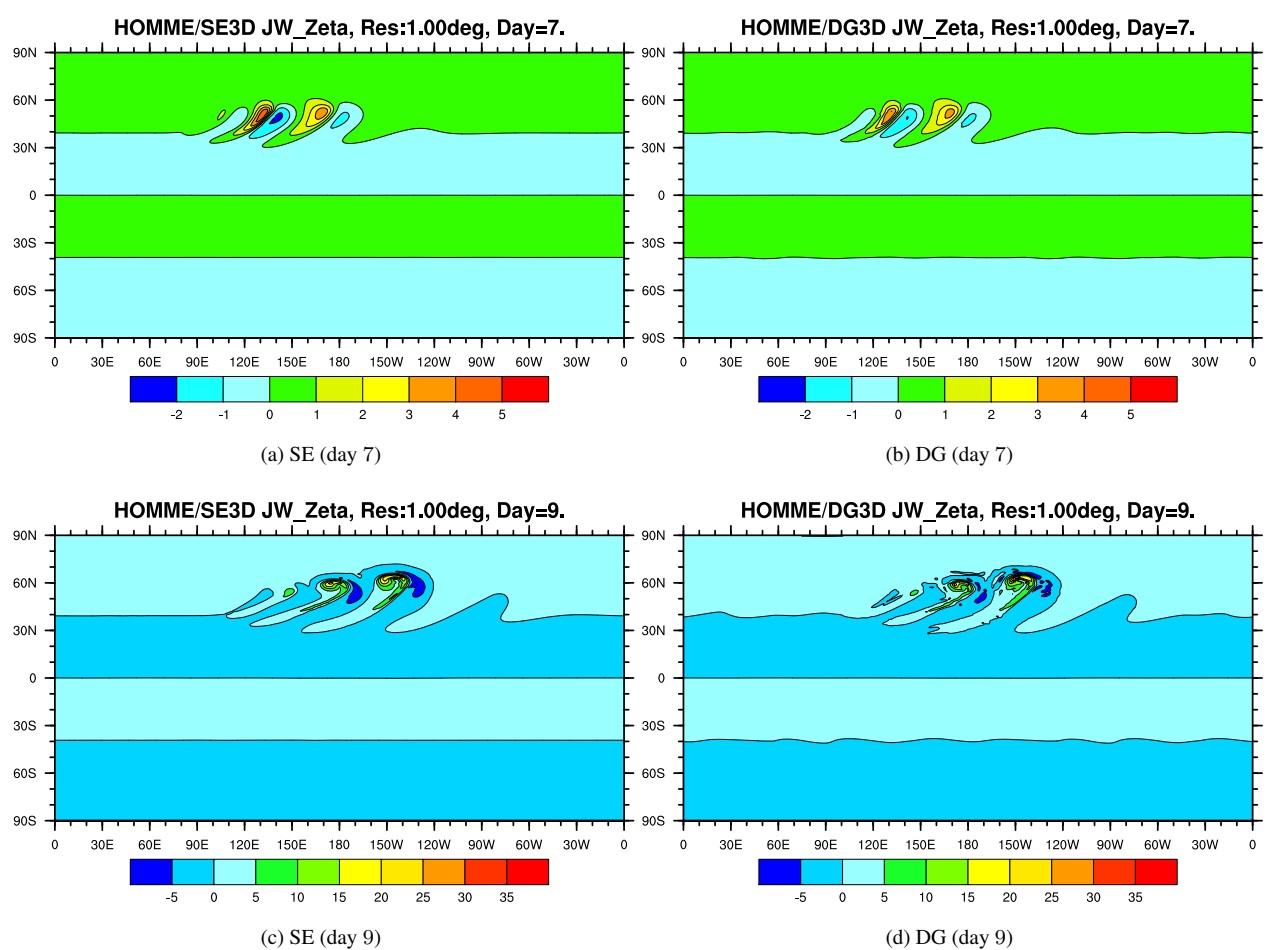

**Figure 4.** Vorticity at day 7 and 9 for the HOMME-SE (a,c) and HOMME-DG (b,d) code for the Jablonowski-Williamson baroclinic wave instability test case. Both methods used 1 degree resolution at the equator ($nlev = 26$, SE: $np = 4$, $n_e = 30$, DG: $np = 6$, $n_e = 18$).

achieved with the old and new communication methods. For the SE method this is not possible due to the varying summation order of the communicated vertex values (see Sec. 5.2).

In the following, we present a series of scaling results to show the effectiveness and performance of our non-blocking asynchronous communication strategy. For both scaling series we use a cubed-sphere mesh with a resolution of $n_e = 60$ and $n_e = 120$ elements along each edge of the cubed-sphere for $E \equiv 21,600$ and $E \equiv 86,400$ total elements respectively.

## 4.2 Scaling results for the SE method

For the SE method, we perform a strong scaling for half-degree $n_e = 60$ and quarter-degree $n_e = 120$ resolutions with $np = 4$ and $nlev = 26$. In order to limit the total amount of computational time, we performed nine days of simulated time for the

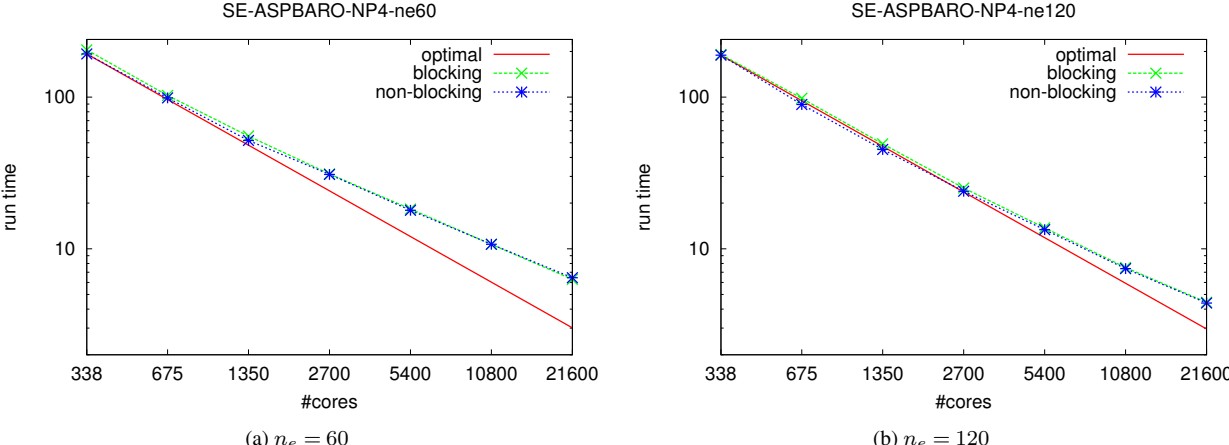

(a) $n_e = 60$                       (b) $n_e = 120$

**Figure 5.** Strong scaling of the SE method in HOMME the Jablonowski-Williamson test baroclinic wave instability test case for $n_e = 60$ (a) and $n_e = 120$ (b). For these runs we used $np = 4$ and $nlev = 26$. For $n_e = 60$ we computed 4860 timesteps and for $n_e = 120$ we computed 1080 timesteps.

$n_e = 60$ runs but only one day of simulated time for the $n_e = 120$ runs. Figure 5 shows the plots of the strong scaling of total time-stepping time for both resolutions. Additionally, Table 1 lists the timing results as well as speed up numbers from the $n_e = 60$ and $n_e = 120$ scaling runs as well.

For moderate numbers of elements per process, we see a significant decrease in run time when using asynchronous communication. However, once the number of elements per process decreases below four elements per process, the advantage of using asynchronous communication becomes negligible. This is due to the fact that there is a smaller amount of interior packing and unpacking to be done while the messages are being sent and received.

Finally, for $n_e = 120$ on 338 processes we see that there is a negligible performance improvement (1.009x) when using the asynchronous method. Here, the movement of element edge and vertex data is a large part of the total run time. Although this data movement hides some of the communication costs, the decrease in memory locality when packing/unpacking individual messages compared to packing/unpacking entire elements can increase the total cost of data movement relative to the original communication method.

### 4.3 Scaling results for the DG method

For the DG ($np = 6$, $nlev = 26$) strong scaling we compare four different communication methods. The pre-existing method using the same connectivity as the SE method (see Figure 2a) is referred to synchronous. The method implementing asynchronous communication but with the SE connectivity is called overlap (vx). The remaining two methods use the reduced connectivity described in Figure 2b. One method only uses the overlapping of pack/unpack with send/receive and is referred to as asynchronous. The other method uses the overlapping of computation as described in Algorithm 3 and is simply denoted overlapping.

**Table 1.** Results for the strong scaling of the SE method ($np = 4$ and $nlev = 26$) and for $n_e = 60$ (a) nd $n_e = 120$ (b). We list the number processes P, the maximum number of elements per process $E/P$, and the times for the synchronous and asynchronous communication methods. The speed up of using asynchronous communication is included in parentheses.

(a) $n_e = 60$

| $P$ | $E/P$ | synch. | asynch. |
|---|---|---|---|
| 338 | 64 | 204.64 | 192.518 (1.063) |
| 675 | 32 | 102.50 | 98.85 (1.037) |
| 1350 | 16 | 55.32 | 51.78 (1.068) |
| 2700 | 8 | 31.16 | 30.93 (1.007) |
| 5400 | 4 | 18.29 | 17.94 (1.020) |
| 10800 | 2 | 10.69 | 10.71 (0.998) |
| 21600 | 1 | 6.29 | 6.45 (0.975) |

(b) $n_e = 120$

| $P$ | $E/P$ | synch. | asynch. |
|---|---|---|---|
| 338 | 256 | 190.90 | 189.108 (1.009) |
| 675 | 128 | 98.14 | 89.43 (1.097) |
| 1350 | 64 | 49.19 | 45.18 (1.089) |
| 2700 | 32 | 25.15 | 23.97 (1.049) |
| 5400 | 16 | 13.73 | 13.37 (1.027) |
| 10800 | 8 | 7.52 | 7.40 (1.017) |
| 21600 | 4 | 4.44 | 4.39 (1.011) |

In Figure 6 the strong scaling results for the DG code ($np = 6$, $nlev = 26$) for Jablonowski-Williamson test case are presented. The numbers used to generate the plots are presented in Table 2. We can see that the using the asynchronous communication leads to improved performance. Here, we encounter a performance gain of approximately $8\%$. This is increased by reducing the connectivity to over $10\%$ which is not surprising. More interesting is, as the numbers for the non-blocking and the overlapping runs show, that placing some work (other than the pack/unpack) between the send and receive calls increases the overall performance of the simulation even further. This is a strong indicator that a code revision (HOMME was originally only designed and optimized for SE) such that the maximum amount of computation can be placed between the send and receive calls will be beneficial.

## 5 Discussion

### 5.1 Performance at Large-Scale

As seen in Section 4.2 the non-blocking asynchronous communication method yields significant performance increases when the number of elements per MPI process is four or above. This is due, in part, to the limited amount of data associated with element boundaries when there are few elements per process. Thus this technique is mainly beneficial when there are a moderate number of elements per MPI process. Although HOMME scales fairly well out to one element per MPI process, production climate runs typically assign more elements per process (Small et al. (2014)). In this regime, the asynchronous communication scheme is significantly more efficient.

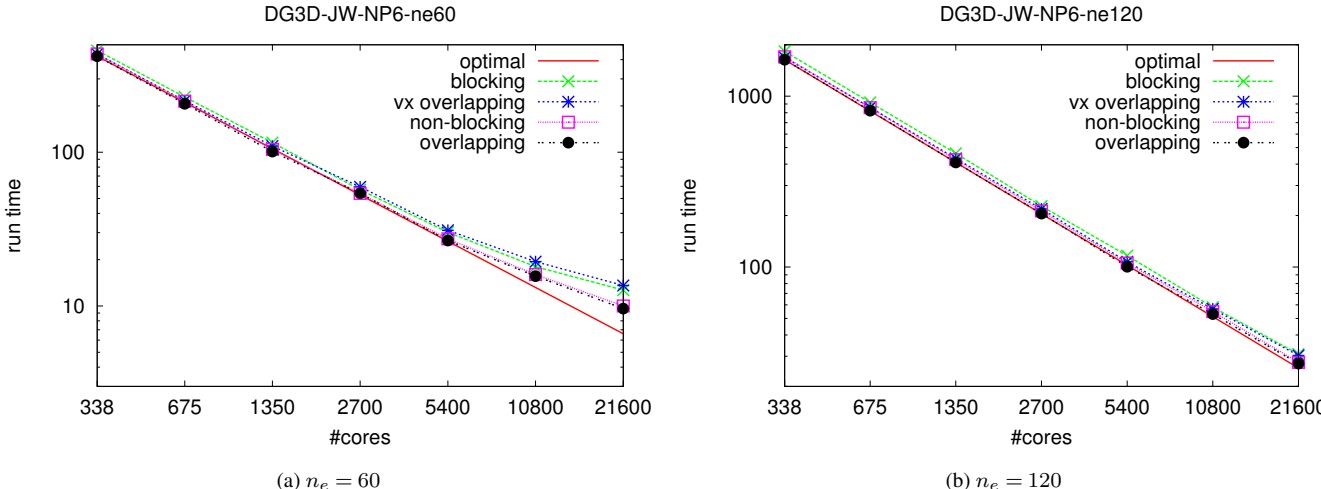

**Figure 6.** Strong scaling of the HOMME-DG code for the Jablonowski-Williamson baroclinic wave instability test case. For the run we used $np = 6$, $nlev = 26$, and (a) $n_e = 60$ as well as (b) $n_e = 120$. For each run we compute 4500 timesteps.

## 5.2 Bit-for-Bit Reproducibility

In the non-blocking asynchronous communication methods, messages received from other processes are additively unpacked as they are received through the use of MPI_Testany. Due to the indeterminate ordering of these contributions and the fact that finite precision floating point arithmetic is non-associative, two identical runs, which may have MPI messages received in different orders, will not produce the exact same, bit-for-bit, results. Although a stable numerical scheme will produce qualitatively similar results, quantitative differences may be present.

This complexity can confound traditional methods of verifying the correctness of simulations, ports to other machines, or code changes. However, knowledge of the numerical accuracy of the underlying integration and discretization schemes can be used to bound this difference and restore confidence in the accuracy of the dynamic results. Additionally, statistical techniques such as (Baker et al. (2015)) can be used to verify that the differences are limited to machine level round-off and will not have a drastic impact on qualitative results.

Although techniques such as Kahan summation (Kahan (1965)) can limit the amount of accumulated machine precision round-off error, ensuring bit-for-bit exactness between identical runs requires more care. One possible avenue would be to unpack messages as they are received storing this data in another buffer and waiting to perform additive operations until all messages have been received. This enforces a static order of operations and avoids the differences caused by non-associativity.

**Table 2.** Time in seconds for the synchronous, the overlapping with vertex connectivity, the asynchronous without vertex connectivity, and the overlapping without vertex connectivity communication methods for the DG ($np = 6$, $nlev = 26$) strong scaling with for $n_e = 60$ (a) and $n_e = 120$ (b). $P$ denotes the number of cores used in the simulation.

(a) $n_e = 60$

| $P$ | $E/P$ | synch. | overlap(vx) | asynchronous | overlapping |
|---|---|---|---|---|---|
| 338 | 63.9 | 458.88 | 434.73 (1.056) | 435.52 (1.054) | 421.32 (1.089) |
| 675 | 32 | 228.54 | 216.37 (1.056) | 214.45 (1.066) | 206.68 (1.106) |
| 1350 | 16 | 115.44 | 109.64 (1.053) | 104.45 (1.105) | 101.02 (1.143) |
| 2700 | 8 | 56.95 | 59.48 (0.957) | 53.95 (1.056) | 54.12 (1.052) |
| 5400 | 4 | 30.15 | 31.07 (0.970) | 27.32 (1.103) | 26.63 (1.132) |
| 10800 | 2 | 18.02 | 19.42 (0.928) | 16.00 (1.126) | 15.65 (1.152) |
| 21600 | 1 | 12.69 | 13.58 (0.934) | 10.00 (1.269) | 9.62 (1.318) |

(b) $n_e = 120$

| $P$ | $E/P$ | synch. | overlap(vx) | asynchronous | overlapping |
|---|---|---|---|---|---|
| 338 | 255.6 | 1829.71 | 1686.99 (1.085) | 1700.86 (1.076) | 1635.55 (1.119) |
| 675 | 128 | 917.91 | 859.39 (1.068) | 856.77 (1.071) | 822.08 (1.117) |
| 1350 | 64 | 462.95 | 432.00 (1.072) | 424.79 (1.090) | 409.40 (1.131) |
| 2700 | 32 | 227.26 | 218.71 (1.039) | 212.71 (1.068) | 205.41 (1.106) |
| 5400 | 16 | 116.40 | 107.28 (1.085) | 105.41 (1.104) | 100.39 (1.159) |
| 10800 | 8 | 58.18 | 56.80 (1.024) | 54.82 (1.061) | 53.01 (1.097) |
| 21600 | 4 | 30.89 | 30.30 (1.019) | 27.62 (1.118) | 27.19 (1.136) |

## 6  Conclusion

In this paper we outlined our implementation of non-blocking asynchronous communication in HOMME for both the SE and DG methods. This strategy included the use of non-blocking MPI routines as well as a restructuring of the pack and unpack methods to provide data movement as well as other computation during the communication. Most notably, even in the absence of additional computation, the SE method attained performance gains simply by overlapping the packing and unpacking of messages and internal buffers. These gains were most significant when run at a modest number of elements per MPI process, as is typical in production runs.

For the DG method, where additional computation is available to be performed during the communication, there were even bigger efficiency and scalability gains. The scaling results for the DG method also highlighted the increases that could be gained in the SE version if there is additional computation with which to overlap communication.

One limitation of the non-blocking asynchronous communication method, as implemented, is round-off level differences of results between identical runs for the SE method. However, numerical and statistical analysis can be used to bound these differences and restore confidence in simulation results.

We expect that with additional development, non-blocking asynchronous communication will provide more computation overlap, further increasing the performance and scalability of HOMME, CAM, and CESM.

## Code availability and compiler flags

For the test carried out in this study the source code was compiled using the Intel FORTRAN compiler at version 13.1.2 with optimization flags -O3. For the asynchronous communication we set the environment variables MP_EAGER_LIMIT=4194305 and MP_EAGER_LIMIT_LOCAL=4194305.

The source code is available through the **homme_dg_branch** of the HOMME code repository (https://www.homme.ucar. edu/) available in the directory **https://svn-homme-model.cgd.ucar.edu/branches/homme_dg_branch/trunk/src**. The modified and added files[1] are:

**linkage_mod.F90** implementing the linkage pattern described in Figure 2.

**nonblockingcomm_mod.F90** implementing the asynchronous communication described in Algorithm 1 and 2.

**dg3d_packunpack_mod.F90** implementing the pack and unpack routines for the DG method used in Algorithm 1 and 1.

**advect_packunpack_mod.F90** implementing the pack and unpack routines for the SE method used in Algorithm 1 and 1.

## Acknowledgement

We would like to acknowledge high-performance computing support from Yellowstone (Yel) provided by NCAR's Computational and Information Systems Laboratory, sponsored by the National Science Foundation. Robert Klöfkorn acknowledges the DOE BER Program under the award DE-SC0006959, NCAR/CISL's Research and Supercomputing Visitor Program (RSVP) and the Research Council of Norway and the industry partners – ConocoPhillips Skandinavia AS, BP Norge AS, Det Norske Oljeselskap AS, Eni Norge AS, Maersk Oil Norway AS, DONG Energy A/S, Denmark, Statoil Petroleum AS, ENGIE E&P NORGE AS, Lundin Norway AS, Halliburton AS, Schlumberger Norge AS, Wintershall Norge AS – of The National IOR Centre of Norway for financial support.

---

[1]Each file is linked to it's repository location.

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
