# Peer review of "Asynchronous Communication in Spectral Element and Discontinuous Galerkin Methods for Atmospheric Dynamics – A Case Study Using the Higher Order Methods Modeling Environment (HOMME-homme\_dg\_branch)"

_Geoscientific Model Development, 2016_

## Short Comment (SC1) · 30 Mar 2016

Dear authors,

In my role as Executive editor of GMD, I would like to bring to your attention our Editorial version 1.1:

http://www.geosci-model-dev.net/8/3487/2015/gmd-8-3487-2015.html

This highlights some requirements of papers published in GMD, which is also available on the GMD website in the 'Manuscript Types' section:

http://www.geoscientific-model-development.net/submission/manuscript_types.html

[Figure]

In particular, please note that for your paper, the following requirement has not been met in the Discussions paper:

- "The main paper must give the model name and version number (or other unique identifier) in the title."

- "If the model development relates to a single model then the model name and the version number must be included in the title of the paper. If the main intention of an article is to make a general (i.e. model independent) statement about the usefulness of a new development, but the usefulness is shown with the help of one specific model, the model name and version number must be stated in the title. The title could have a form such as, "Title outlining amazing generic advance: a case study with Model XXX (version Y)"."

Please add the information, that you are working with CESM including a version number in the title upon your revised submission to GMD.

Yours,

Astrid Kerkweg

---

## Short Comment (SC2) · 31 Mar 2016

Dear Astrid,

thanks for the hint. I will fix the title accordingly. However, I have on remaining question. There is no explicit version of the source code that we can refer to because the implementation is contained in a development branch. How should I deal with this?

Thanks,

Robert
* * *

---

## Short Comment (SC3) · 31 Mar 2016

Dear Robert,

the best option would be to make the version somehow unambiguously identifiable. Maybe something like: "CESMX.Y_nameandnumberdevbranch" (e.g. CESMX.Y_iris03) where you can freely chose the type of subname and version number of your development branch. If you have good reasons to reject such a naming the very weak statement "based on CESMX.Y" would be also acceptable. But this would not serve the purpose to make the code including your development locatable ....

Cheers, Astrid

---

## Referee Comment (RC1) · Anonymous Referee #1 · 23 Apr 2016

In the paper, the authors describe their implementation of asynchronous communication in in the High-Order Methods Modeling Environment (HOMME) dynamical core of the Community Atmosphere Model (CAM). Data packing is overlapped with communication for the continuous spectral element method in HOMME at the cost of bit-for-bit reproducibility. Thus the Jablonowski–Williamson baroclinic wave instability test case is used to verify the changes to HOMME were accurate. In the discontinuous spectral element method in HOMME, communication is overlapped with data packing as well as computation and the authors were able to achieve bit-for-bit reproducibility. Scaling results on the Yellowstone supercomputer are shown for both the continuous and discontinuous spectral element methods for the the Jablonowski–Williamson baroclinic

wave instability test case.

Investigations on speeding up HOMME, like what is presented in the paper, do have an impact on the community that uses it in CAM and is thus an interesting topic. The paper describes an implementation of overlapping communication and computation that in practice is well know. It focuses on a detail of this that I don't know if it is known. Specifically, posting the send once the data is packed for a single neighbor and not waiting until all the data is packed into the communication buffers. The approach is demonstrated to yield up to a 16% decrease in run time, depending on the number of elements per process. The methods and assumptions are clearly outlined. The authors fail to report on others attempts to overlap communication and computation to support their claim that they their approach is new. With some work, and access to the Yellowstone supercomputer, I believe the authors work is reproducible. They authors provide a subversion repository and branch where a majority of the code can be found.

Although this change to HOMME is incremental, I believe that improvements in model speed are useful for publication. Please find more comments about issues that the authors should address below.

First, I would suggest that the authors proofread the paper again as there are some typographical errors. Below is a small sample of typographical errors and is not meant as an exhaustive list.

1. *typo on page 5:* "eg." should be "e.g."

2. *typo on page 8:* "a auxiliary diagnostic variables" should be "auxiliary diagnostic variables"

Second, I would suggest the authors address the following content concerns.

3. First the authors state :"That is, a process sending a blocking message must wait until the message has been received." This is technically not true if you are

talking about `MPI_Send`. The function `MPI_Send` only blocks until the buffer can be reused. If you are not talking specifically about `MPI_Send` this needs to be clarified.

4. In is unclear how many time-steps were used to compute the numbers in Figure 5 and Tables 1 and 2. Some more detail should be added to the captions.

5. In Figures 1, 5 and 6 line plots are used for discrete data. Is there a piecewise linear fit between the data? I would suggest to use only symbols where the actual data measured is.

6. I believe the title needs to include the code name and version.

7. A literature survey should be given to support the claims that the authors approach is new. I have rarely see this kind of detail presented in the literature, so maybe a review of the most popular finite element methods is in order.

---

## Referee Comment (RC2) · Anonymous Referee #2 · 23 May 2016

This is a short paper describing work done in the finite element boundary communication in the HOMME model, for both CG and DG discretizations. The authors have refactored the communication routines to allow for overlapping of communication and computation. The paper is well written and clearly explained. The speedups are modest, but still significant and the approach should be adopted by the HOMME model. Larger speedups should be possible.

The authors also describe their use of a MPI micro-benchmark for evaluating the effectiveness of non-blocking communication. Results are presented for a particular machine ("yellowstone"). I assume before undertaking the code refactoring necessary to overlap computation and communication, one should establish how effective this can

be on the target machine. What can researchers expect on modern parallel computers? Are the results on yellowstone typical of an average institutional machine?

page 1, typo, "ocaen" -> "ocean"

page 1, line 20, add: "The SE and DG methods attain this property for arbitrary order, at the expense of a small timestep"

―――――――――――――――――――

---

## Referee Comment (RC3) · Anonymous Referee #3 · 25 May 2016

This paper discusses a new non-blocking asynchronous communication for the High-Order Methods Modeling Environment (HOMME) and is therefore very specific to the underlying code and model situation.

The spectral element (SE) dynamical core implemented in HOMME is currently the default dynamical core of the state of the art climate model CESM (atmosphere component CAM). The authors compare two Galerkin schemes, the spectral element method and the discontinuous Galerkin method, on a standard baroclinic wave instability benchmark test. Obviously, their new communication pattern improves the runtime significantly (special for DG) and (maybe even more important) leads to the excellent scalability of HOMME (which was also the case for the synchronous communication).

[Figure]

In general, I recommend the publication, once the authors answer/complete the tasks below.

Comments: - the authors should underline the specific situation in HOMME. In my understanding, HOMME was first designed for SE. Then, a DG dynamical core was implemented. Thus, it is maybe not surprising, that the new communication has a higher impact for the DG version (where the SE communication is naturally suboptimal).

- more literature (besides the specific HOMME publications) should be provided: Is this communication approach already used for/in other (maybe similar) methods/projects? Or is this a novel approach?

-line 1-24: what does it mean beyond 2k cores? Are there no higher scalability results available for DG as for SE?

-Yellowstone: it would be nice, to have more information about this supercomputer (some technical specifications), since runtimes might depend on machines and compiler settings.

-The authors write: more internal vertices provide more data movement and therefore better communication hiding. Since HOMME also has some finite volume schemes implemented, the authors should mention, if their approach would also work for these implementations, since the amount of communication data is much higher.

-line 9-4: ...produce accurate dynamics... I recommend to refer to section 5.2 (see also comment below).

-why is np and ne different for SE and DG? it is not clear to me, which np is used for the performance tests.

-section 5.2.: Knowing that round off errors play an important role, good numerical schemes should be stable with respect to these errors. Thus, I think the bit-for-bit reproducibility is rather a numerical scheme property than a communication issue. The authors could mention this as well, which can be tested with the aid of statistical techniques. In the current version the reader gets the impression that this is an asynchronous communication problem - but in fact we also do not know if the SE solution is right.

-Table 2: something is wrong with the caption description. (b) should be ne=120?

Minor: 1-10: ocean

---

## Author Comment (AC1) · 22 Jun 2016

We thank the reviewer for the helpful and constructive comments.

–

Q1: typo on page 5: "eg." should be "e.g."

A: Fixed.

–

Q2: typo on page 8: "a auxiliary diagnostic variables" should be "auxiliary diagnostic variables"

A: Fixed.

–

Q3: First the authors state :"That is, a process sending a blocking message must wait until the message has been received." This is technically not true if you are talking about MPI_Send. The function MPI_Send only blocks until the buffer can be reused. If you are not talking specifically about MPI_Send this needs to be clarified.

A: The text has been revised accordingly.

–

Q4: In is unclear how many time-steps were used to compute the numbers in Figure 5 and Tables 1 and 2. Some more detail should be added to the captions.

A: Fixed.

–

Q5: In Figures 1, 5 and 6 line plots are used for discrete data. Is there a piecewise linear fit between the data? I would suggest to use only symbols where the actual data measured is.

A: We prefer to show both, the data points and the general trend (by connecting the data points).

–

Q6: I believe the title needs to include the code name and version.

A: Fixed. HOMME and homme_dg_branch was added.

–

Q7: A literature survey should be given to support the claims that the authors approach is new. I have rarely see this kind of detail presented in the literature, so maybe a review

of the most popular finite element methods is in order.

A: We have added a survey of both, dynamical cores for NWP, e.g. NUMA, ICON, MPAS-A, and NICAM as well as other contemporary simulation software presented for the prestigious Gordon Bell price as part of the International Conference on High Performance Computing, Networking, Storage and Analysis.

———————————————

---

## Author Comment (AC2) · 22 Jun 2016

We thank the reviewer for the helpful and constructive comments.

–

Q: What can researchers expect on modern parallel computers?

A: The topic of asynchronism is widely discussed in the HPC community (cf. Exaflop/s: The why and the how, D.E.Keyes, 2011) and will definitely play an important role on future machines. Thus, we believe that implementing techniques that improve asynchronism in current simulations models will be beneficial in the future. I have added the citation to the paper.

—

Q: Are the results on yellowstone typical of an average institutional machine?

A: As pointed out in the paper "Asynchronous MPI for the Masses" by Wittmann et al. (2013) "Depending on the implementation quality of the [MPI] library the overhead ranges from negligible to large". This paper gives a good overview on capabilities of different MPI library implementations in context of asynchronous communication and I have added it to the references. However, MPI implementations hopefully improve in the future and thus on-site tests using the Sandia MPI micro benchmark are recommended. In addition, the slower the network the better the scaling when employing asynchronous communication. In that sense we expect the benefits to be large at lower end parallel computers.

—

Q: page 1, typo, "ocaen" -> "ocean"

A: Fixed.

—

Q: page 1, line 20, add: "The SE and DG methods attain this property for arbitrary order, at the expense of a small timestep"

A: We added "..., at the expense of a smaller timestep"

---

## Author Comment (AC3) · 22 Jun 2016

We thank the reviewer for the helpful and constructive comments.

–

Q: The authors should underline the specific situation in HOMME. In my understanding, HOMME was first designed for SE. Then, a DG dynamical core was implemented. Thus, it is maybe not surprising, that the new communication has a higher impact for the DG version (where the SE communication is naturally suboptimal).

A: We agree. While the DG method has lower connectivity which is beneficial for scalability it also suffers from a more severe time step restriction. However, our results

show that (in the DG case) more computation between send and receive increases the performance. We have revised section 4.3 accordingly.

–

Q: More literature (besides the specific HOMME publications) should be provided: Is this communication approach already used for/in other (maybe similar) methods/projects? Or is this a novel approach?

A: We have added a survey of both, dynamical cores for NWP, e.g. NUMA, ICON, MPAS-A, and NICAM as well as other contemporary simulation software presented for the prestigious Gordon Bell price as part of the International Conference on High Performance Computing, Networking, Storage and Analysis.

–

Q: line 1-24: what does it mean beyond 2k cores? Are there no higher scalability results available for DG as for SE?

A: We added a reference to another work showing scalability of both methods beyond the numbers presented here.

–

Q: Yellowstone: it would be nice, to have more information about this supercomputer (some technical specifications), since runtimes might depend on machines and compiler settings.

A: The details on Yellowstone are available through the permanent link provided in the references. Compiler version and flags have been added to the code section.

–

Q: The authors write: more internal vertices provide more data movement and therefore better communication hiding. Since HOMME also has some finite volume schemes

implemented, the authors should mention, if their approach would also work for these implementations, since the amount of communication data is much higher.

A: The approach is applicable to any point-to-point communication. An appropriate sentence was added.

–

Q: line 9-4: ...produce accurate dynamics... I recommend to refer to section 5.2 (see also comment below).

A: The sentence has been changed to "reproduce the results obtained with the pre-existing communication strategy".

–

Q: why is np and ne different for SE and DG? it is not clear to me, which np is used for the performance tests.

A: Throughout the paper we use np=4 for SE (the default also in CAM-SE) and np=6 for DG because for lower np the DG method is not stable for the baroclinic test case due to missing limiter or filter methods.

–

Q: section 5.2.: Knowing that round off errors play an important role, good numerical schemes should be stable with respect to these errors. Thus, I think the bit-for-bit reproducibility is rather a numerical scheme property than a communication issue. The authors could mention this as well, which can be tested with the aid of statistical techniques. In the current version the reader gets the impression that this is an asynchronous communication problem - but in fact we also do not know if the SE solution is right.

A: We have added a reference to the work of Baker et al. (2015) where exactly this issue is addressed in the context of CESM. The results produced by the SE method

using the new communication are correct within the accepted norms. The bit-for-bit reproducibility is a to strict measure in this case.

–

Q: Table 2: something is wrong with the caption description. (b) should be ne=120?

A: Fixed.

–

Q: Minor: 1-10: ocean

A: Fixed.

---

## Author Comment (AC5) · 22 Jun 2016

Dear Astrid,

we have addressed all comments of the reviewers and revised the paper . I have set you a revised version of the paper. Please let me know if additional information is needed.

With best regards,

Robert
* * *

---

## Author Response (AR2)

We thank the reviewer for the helpful and clarifying comments.

——————

Q: in section 4.3. write the statement clearer. ....a code revision (HOMME was originally only designed and optimized for SE).... and even mention this in the introduction (p. 1 line 16)

A: We added the necessary lines to the Introduction and Section 4.3.

——————

Q: The authors write: more internal vertices provide more data movement and therefore better communication hiding. Since HOMME also has some finite volume schemes implemented, the authors should mention, if their approach would also work for these implementations, since the amount of communication data is much higher. -I can not find a clear answer in the text. Obviously the authors did not test it.

A: We did not test the FV schemes in HOMME with the new strategy. However, the approach is easily to be extended to these schemes since the concept of communication is the same just the pack and unpack routines differ due to different data structures used in the FV methods. To extend the approach one would need to write a routine to establish the linkage between processes (by using the old communication method for example) and then a pack and unpack method needs to be provided accounting for the pack/unpack by communication link. These new pack/unpack routines should look very similar to the old ones and we provided two examples on how this should look like. The implemented new communication methods use function pointers to the pack/unpack routines, so no change needed here. We have added a passage in the text and a citation to one of the papers by Erath et al. discussing FV methods in HOMME. Communication of larger amounts of data also means packing and unpacking of larger amounts of data so there should be some benefit using the suggested methodology. Whether the method allows for further overlapping of computation and communication should be addressed in studies subsequent to this work.

——————

Q: why is np and ne different for SE and DG? it is not clear to me, which np is used for the performance tests. -I can not find a clear answer in the text. Using different ne and np affects the working load for each core (compare Fig 5(b) and Fig 6(b))? Getting scalability for DG "is easier"???

A: Fixed. np and nlev have been added to places where either method is mentioned. For the DG method np=6 had to be used otherwise the method shows instabilities for the particular test case used in this paper. Investigations on the stability of the DG methods for certain test cases is, however, not the subject of this paper. For the SE method, on the other hand, it made sense to use np=4 since that is the default configuration used in climate simulations. Furthermore, we did not compare the scalability of both methods and therefore we don't see a problem in using different np for SE and DG. Conclusions about improvements were only drawn for each method separately. However, since the original implementation of the DG method uses exactly the same communication as the SE method, we still can conclude that the proposed changes would be very likely to also have a positive effect for the SE method and thus justify a certain amount of work put into code restructuring.

[revised manuscript text omitted]